# Antiarrhythmic Effects of Supercritical Extract of *Acmella oleracea* in Rats: Electrophysiological Evidence and Cardioprotective Potential

**DOI:** 10.3390/plants14182848

**Published:** 2025-09-12

**Authors:** Ana Paula de Souza e Silva, Flávia Cristina Seabra Pires, Maria Caroline Rodrigues Ferreira, Letícia Maria Martins Siqueira, Eduardo Gama Ortiz de Menezes, Maria Eduarda Ferraz de Carvalho, Luis Adriano Santos do Nascimento, Alberdan Silva Santos, Akira Otake Hamoy, Moisés Hamoy, Raul Nunes de Carvalho

**Affiliations:** 1LABTECS (Supercritical Technology Laboratory), PPGCTA (Graduate Program in Food Science and Technology), ITEC (Institute of Technology), UFPA (Federal University of Pará), Augusto Correa Street S/N, Guamá, Belém 66075-900, PA, Brazil; apdesouzaesilva@gmail.com (A.P.d.S.e.S.); flaviapiress@gmail.com (F.C.S.P.); carolinerof@gmail.com (M.C.R.F.); 2LABTECS (Supercritical Technology Laboratory), PRODERNA (Graduate Program in Engineering of Natural Resources of the Amazon), UFPA (Federal University of Pará), Augusto Corrêa Street S/N, Belém 66075-900, PA, Brazil; leticiammsiqueira@outlook.pt; 3LABTECS (Supercritical Technology Laboratory), Federal Institute of Education, Science and Technology of Rondônia, IFRO, Porto Velho 78900-000, RO, Brazil; ortizegom@hotmail.com; 4Socio Environmental and Water Resources Institute (ISARH), Faculty of Biology, UFRA (Rural Federal University of the Amazon), President Tancredo Neves Avenue, Belém 66077-830, PA, Brazil; maria.carvalho@discente.ufra.edu.br; 5Institute of Biological Sciences, Faculty of Biotechnology, UFPA (Federal University of Pará), Augusto Corrêa Street S/N, Belém 66075-900, PA, Brazil; adriansantos@ufpa.br (L.A.S.d.N.); alberdan@ufpa.br (A.S.S.); 6Institute of Biological Sciences, UFPA (Federal University of Pará), Augusto Corrêa Street S/N, Belém 66075-900, PA, Brazil; hamoyufpa@gmail.com (A.O.H.); hamoy@ufpa.br (M.H.); 7Institute of Technology, Faculty of Food Engineering, UFPA (Federal University of Pará), Augusto Corrêa Street S/N, Belém 66075-900, PA, Brazil

**Keywords:** *Acmella oleracea*, supercritical extraction, spilanthol, cardiac electrophysiology, antiarrhythmic activity, rat model, natural products, cardiovascular pharmacology

## Abstract

Cardiovascular diseases (CVDs) are the leading cause of death worldwide, with cardiac arrhythmias being one of the main factors contributing to morbidity. Currently, several established antiarrhythmic medications with proven efficacy are available. However, frequent use of these medications causes adverse effects with medium- and long-term use. This necessitates the development of new medications, preferably of natural origin and with ethnopharmacological relevance. In this sense, *Acmella oleracea* presents itself as an alternative for the treatment of arrhythmia, considering studies suggesting its cardioprotective effect. Therefore, the objective of this study was to evaluate the electrophysiological and antiarrhythmic effects of a supercritical extract of *Acmella oleracea* (SEAO) in rats. The extract was obtained by supercritical CO_2_ extraction at 70 °C and 320 bar, with an extract yield of 9.72 ± 0.26% (db) and a spilanthol yield of 25.91%. The extract was administered intraperitoneally at doses of 10, 15, and 20 mg/kg in two experimental models: (1) assessment of cardiac electrophysiology and (2) epinephrine-induced arrhythmia. Electrocardiogram (ECG) parameters were measured and compared with controls treated with epinephrine and lidocaine. The SEAO group maintained sinus rhythm and preserved cardiac intervals, with a significant reduction in heart rate and R-R interval compared to the epinephrine group. These findings demonstrate that SEAO exerts dose-dependent antiarrhythmic effects comparable to those of lidocaine. The results corroborate the potential use of SEAO as a natural alternative for arrhythmia management, encouraging further pharmacological and clinical studies.

## 1. Introduction

Cardiovascular diseases (CVDs) encompass all diseases of the heart and blood vessels and are the most common cause of death worldwide. The highest mortality rates are found in Africa, Asia, Eastern Europe, and South America [1], largely due to risk factors such as smoking, alcohol consumption, poor diet, air pollution, population aging, and limited implementation of public policies [2].

According to the World Health Organization (WHO), these diseases, classified as non-communicable, are responsible for more than 17 million deaths annually, with heart attacks and strokes predominating and increasingly affecting younger people [3]. Among the clinical manifestations of CVDs, cardiac arrhythmias are a major cause of morbidity and sudden death, due to disorganization of the heart rhythm and consequent impairment of the heart’s contractile function, which leads to heart attacks and strokes [4]. Although antiarrhythmic drugs are used as therapeutic options, several challenges related to these solutions need to be addressed and should be considered in future drug discovery. Administration challenges, including their narrow therapeutic window and adverse reactions, are among the challenges to be addressed [5].

Several antiarrhythmic drugs are commonly used in the treatment of arrhythmias, and these originate from medicinal plants. This is the case with digoxin, amiodarone, atropine, and lidocaine, for example [6]. However, these agents frequently cause side effects and limit prolonged use, in addition to having variable efficacy depending on the etiology and type of arrhythmia. Therefore, the development of safer and more effective therapies is necessary, especially from natural sources with a history of ethnopharmacological application.

Medicinal plants have been widely studied as potential sources of bioactives. One example is *Acmella oleracea* (L.) R.K. Jansen, popularly known as jambu, an Amazonian species traditionally used in food and folk medicine as an analgesic and anti-inflammatory agent.

Studies have shown that *A. oleracea* extract contains bioactive compounds with antioxidant [7,8,9], anti-inflammatory [10,11,12], hypotensive, vasorelaxant, and diuretic effects [13,14,15], notably spilanthol, an alkamide with extensive pharmacological activity. The extract’s vasorelaxant and diuretic actions suggest a possible effect on the renin–angiotensin–aldosterone system, inhibiting the angiotensin-converting enzyme (ACE) and resulting in cardioprotective effects.

Despite these previously described properties, no studies have evaluated the effects of jambu on cardiac electrophysiological parameters, especially regarding its potential for arrhythmia control. Furthermore, although jambu and spilanthol in particular are classified as safe, the method of extraction can influence their toxicity level and, in some cases, render them unsuitable for use. Considering this, supercritical extraction using carbon dioxide (CO_2_) as a solvent allows for the production of concentrated extracts free of toxic solvents and rich in compounds such as spilanthol. As previously demonstrated in the literature, it is the most efficient method for obtaining a concentrated extract containing spilanthol [16], favoring the development of safe and standardized plant-derived protection products. Furthermore, CO_2_ is recognized as “Generally Recognized as Safe” (GRAS), which further guarantees its effectiveness and safety [17].

Given this scenario, the present study aimed to investigate the effects of a supercritical extract of *Acmella oleracea* (SEAO) on cardiac electrophysiological function and antiarrhythmic activity in an animal model.

## 2. Results

### 2.1. Extraction Yield and Extract Analysis

The supercritical extraction process at 70 °C and 320 bar resulted in an overall extract yield of 9.72 ± 0.26% (db). A study evaluating *Acmella oleracea* flowers using the supercritical extraction method reported yield values that are consistent with those obtained in this study. Dias et al. (2017) [18] obtained yields ranging from 7 to 9%, with the 9% yield obtained using the same operating conditions as in this study, confirming the precision of the process.

Figure 1 shows the extract chromatogram. Peak 17.41 was identified as spilanthol, confirming the efficiency of the process for extracting this compound. The retention time of the spilanthol peak was 17.41 min. The extract presented a chromatographic area of 4.49 × 10^10^ µA.s, corresponding to 25.91% of the total chromatogram area, indicating a significant concentration of spilanthol in the extract.

The quantification of spilanthol in *Acmella oleracea* flower extracts obtained by supercritical extraction has been reported in the literature. Dias et al. (2012) [16] obtained a spilanthol yield of 65.4%. Dias et al. (2017) [18] obtained yields ranging from 0.57% to 2.60% for different operating conditions. Barbosa et al. (2017) [19] obtained spilanthol yields ranging from 0.31% to 1.07% for different drying and storage methods of *Acmella oleracea*. The percentage obtained in this work demonstrates the efficiency of the supercritical extraction process, since it was possible to obtain a higher percentage than most of those reported in the literature.

### 2.2. SEAO Electrocardiographic Activity

This study analyzed the P-Q interval, which represents the time interval between atrial contraction and the onset of ventricular contraction. The duration of the QRS complex, which represents the time for ventricular contraction, was analyzed. The Q-T interval, which represents the entire ventricular cycle between contraction and repolarization, was also analyzed. The amplitude, which represents the signal intensity in mV captured by the electrode, was analyzed. The R-R interval was evaluated, which represents the heart rhythm and its variation may indicate arrhythmias. The results are summarized in Table 1.

The negative control group presented normal cardiac rhythm, with amplitude above 1.0 mV (Figure 2A), a QRS complex preceded by a P wave (Figure 2B), and a normal R-R interval (Figure 2C).

The stimulant positive control showed increased cardiac activity (Figure 3A) and the presence of a P wave before the QRS complex, characterizing sinus rhythm (Figure 3B). However, there was variation in the ECG in the QRS amplitude (mV) and in the P wave (Figure 3B,C), which was expected, since epinephrine already has proven cardiac activity-enhancing effects.

The inhibitory positive control showed similarity to the recordings obtained from the negative control group (Figure 3D–F), showing sinus rhythm and regular cardiac function, which was also expected, since lidocaine is already frequently used to treat malignant arrhythmias [18].

For the evaluated groups that received SEAO at doses of 10, 15, and 20 mg/kg i.p., all records showed sinus rhythm, and there were no changes in heart rhythm, as they presented ECGs comparable to those of the inhibitory positive control group (Figure 4).

To evaluate the statistical differences between the groups, the means of frequencies (bpm), amplitude (mV), R-R interval (ms), P-Q interval (ms), QRS complex duration, and Q-T interval were compared regarding the effects on cardiac electrophysiology. Regarding heart rate, the negative control group had a mean of 266.7 ± 9.849 bpm, showing a statistical difference compared with the positive control stimulant group, which had a mean of 369.3 ± 38.94 bpm. The heart rate for the positive control inhibitory group showed a statistical difference in relation to the positive control stimulant group. The SEAO groups had means of 249.3 ± 7.348 bpm, 250.2 ± 5.426 bpm, and 248.9 ± 8.069 bpm for the doses of 10, 15, and 20 mg/kg i.p., respectively, showing a statistical difference in relation to the stimulant positive control group (F(5.48) = 62.10; *p* < 0.001) (Figure 5A).

The amplitude showed no statistical difference in the ECG evaluation. The negative control group had a mean of 1.299 ± 0.1027 mV, the stimulant positive control group had a mean of 1.196 ± 0.1105 mV, the inhibitory positive control group had a mean of 1.302 ± 0.1415 mV, and the 10, 15, and 20 mg/kg i.p. SEAO groups had means of 1.376 ± 0.1381 mV, 1.316 ± 0.1625 mV, and 1.238 ± 0.1098 mV, respectively (F(5.48) = 2.130; *p* = 0.0778) (Figure 5B).

For the R-R interval, the negative control group had a mean of 224.4 ± 13.15 ms, demonstrating a statistical difference from the stimulant positive control group (156.6 ± 2.963 ms). The inhibitory positive control group had a mean of 234.6 ± 14.52 ms, showing a statistical difference from the stimulant positive control group. The SEAO 10.15 and 20 mg/kg groups had respective means of 239.1 ± 7.541 ms, 242.1 ± 9.610 ms, and 243.2 ± 11.88 ms, showing no differences from the inhibitory positive control groups, but showing a statistical difference from the negative and stimulant positive control groups (F(5.48) = 88.31; *p* < 0.0001) (Figure 6A). For the P-Q interval that corresponds to atrial activation and the onset of ventricular activity, the negative control group had a mean of 43.30 ± 1.855 ms, demonstrating a statistical difference from the stimulant positive control group (32.14 ± 1.527 ms). The inhibitory positive control group had a mean of 43.93 ± 1.371 ms, with a statistical difference from the stimulant positive control group. For the SEAO groups, the group that received 10 mg/kg had a mean of 43.53 ± 1.834 ms, showing a statistical difference from the stimulant positive control group. The group that received 15 mg/kg had a mean of 48.09 ± 5.146 ms and showed a statistical difference from all other groups in the study, with the exception of the SEAO 20 mg/kg group. For the group that received SEAO at a dose of 20 mg/kg, the mean was 52.24 ± 4.134 ms, showing a statistical difference from the negative control, stimulant positive control, and inhibitory positive control groups and the SEAO 10 mg/kg group (F(5.48) = 44.69; *p* < 0.001) (Figure 6B).

When analyzing the duration of the QRS complex, the negative control group had a mean of 11.80 ± 1.507 ms, demonstrating a statistical difference from the positive stimulant control group, which presented a mean QRS duration of 6.744 ± 1.039 ms. The positive inhibitory control group presented a mean of 11.63 ± 1.420 ms, with a statistical difference from the positive stimulant control group. The SEAO groups that received doses of 10, 15, and 20 mg/kg differed statistically from the positive stimulant control group (F(5.48) = 17.43; *p* < 0.0001) (Figure 7A). For the Q-T interval, which corresponds to the ventricular cycle from the beginning of depolarization to repolarization, the negative control group had a mean of 24.31 ± 1.199 ms, showing a statistical difference from the stimulant positive control group, which had a mean of 17.99 ± 1.037 ms. The inhibitory positive control group had a mean of 22.26 ± 1.928 ms, which showed a statistical difference from the inhibitory positive control. The SEAO groups of 10, 15, and 20 mg/kg had respective means of 23.48 ± 1.359 ms, 22.69 ± 1.941 ms, and 22.11 ± 2.494 ms, which showed a statistical difference from the stimulant positive control group (F(5.48) = 14.38; *p* < 0.001) (Figure 7B).

### 2.3. Antiarrhythmic Activity of SEAO

The positive control and SEAO 10, 15, and 20 mg/kg i.p. groups showed maintenance of the electrophysiological characteristics of the heart, but with an increase in frequency for the positive control, SEAO 10 mg/kg, and SEAO 15 mg/kg groups, and a reduction in frequency for the SEAO 20 mg/kg group (Figure 8 and Figure 9). Table 2 presents the results obtained for the evaluated antiarrhythmic activity.

After administration of epinephrine in the positive control and SEAO 10, 15, and 20 mg/kg i.p. groups, the heart rate of the control group averaged 266.7 ± 9.849 bpm, and the epinephrine group (described in Section 2.2 as the stimulant positive control) averaged 369.3 ± 38.94 bpm. The heart rate for the group that received lidocaine + epinephrine averaged 323.3 ± 18.08 bpm, showing a statistical difference from the control and epinephrine groups. The groups that received 10 and 15 mg/kg i.p. of SEAO + epinephrine, respectively, averaged 327.3 ± 11.00 bpm and 323.8 ± 17.47 bpm, showing a statistical difference from the control and epinephrine groups. The group receiving the 20 mg/kg i.p. dose of SEAO had a mean heart rate of 298.0 ± 22.76 bpm, with a statistically significant difference from the control, epinephrine, and SEAO 10 mg/kg i.p. groups (F(5.48) = 21.79; *p* < 0.001) (Figure 10A).

For the amplitude assessment, there was no statistical difference between the groups that received prior treatment with lidocaine or different doses of SEAO. The control group had a mean of 1.299 ± 0.1027 mV, the epinephrine group had a mean of 1.196 ± 0.1105 mV, the lidocaine + epinephrine group had a mean of 1.251 ± 0.05730 mV, and the SEAO 10 mg/kg i.p. group (1.201 ± 0.1285 mV), the SEAO 15 mg/kg i.p. group (1.191 ± 0.1385 mV), and the SEAO 20 mg/kg i.p. group (1.158 ± 0.04980 mV) (F (5.48) = 2.134; *p* = 0.0774) (Figure 10B).

For analysis of the R-R interval, the control group had a mean of 224.4 ± 13.15 ms, and the epinephrine group had a mean R-R interval of 156.6 ± 2.963 ms. The group that received lidocaine + epinephrine had a mean R-R interval of 192.7 ± 19.44 ms, showing a statistical difference from the control and epinephrine groups. The group that received SEAO 10 mg/kg + epinephrine, which had a mean of 184.8 ± 18.61 ms, maintained a statistical difference from the control and epinephrine groups. The group that received SEAO 15 mg/kg i.p. had a mean of 205.1 ± 8.388 ms, with differences from the control, epinephrine, and SEAO 10 mg/kg groups. The SEAO 20 mg/kg + epinephrine group had a mean of 230.7 ± 9.785 ms, which was statistically different from the epinephrine, lidocaine, SEAO 10 mg/kg, and SEAO 15 mg/kg groups (F(5.48) = 37.57; *p* < 0.0001) (Figure 11A).

For the P-Q interval, the control group had a mean of 43.30 ± 1.855 ms, and the epinephrine group had a mean of 32.14 ± 1.527 ms. The P-Q interval for the lidocaine + epinephrine group was 52.23 ± 2.121 ms, which was different from the control and epinephrine groups. The groups that received SEAO at doses of 10 and 15 mg/kg had respective means of 57.06 ± 5.509 ms and 57.41 ± 4.229 ms, showing statistical differences from the control, epinephrine, and lidocaine groups. The group that received 20 mg/kg i.p. of SEAO had a mean of 61.74 ± 4.452 ms and showed a statistical difference from the control, epinephrine, lidocaine, and SEAO 10 mg/kg i.p. and SEAO 15 mg/kg groups (F(5.48) = 83.99; *p* < 0.001) (Figure 11B).

In the analysis of the duration of the QRS complex, the control group had a mean of 11.80 ± 1.507 ms; the epinephrine group had a mean QRS duration of 6.744 ± 1.039 ms. The lidocaine + epinephrine group had a mean of 11.11 ± 2.147 ms, showing a statistical difference from the epinephrine group. The group that received SEAO at a dose of 10 mg/kg i.p. + epinephrine showed a statistical difference from the control and epinephrine groups. The groups receiving 15 mg/kg and 20 mg/kg of SEAO + epinephrine, respectively, had means of 11.93 ± 1.521 ms and 12.72 ± 0.4790 ms, which showed a statistical difference from the control, epinephrine, and SEAO 10 mg/kg groups (F(5.48) = 22.92; *p* < 0.0001) (Figure 12A).

For the Q-T interval, the control group had a mean of 24.31 ± 1.199 ms, and the epinephrine group had a mean of 17.99 ± 1.037 ms. The group receiving lidocaine + epinephrine had a mean of 23.01 ± 1.364 ms, which showed a difference from the epinephrine group. The group receiving 10 mg/kg i.p. of SEAO had a mean of 20.64 ± 1.365 ms, demonstrating a statistical difference from the control, epinephrine, and lidocaine groups. For the groups that received 15 and 20 mg/kg i.p., the respective means were 21.88 ± 1.637 ms and 22.26 ± 1.477 ms, demonstrating statistical differences from the control, epinephrine, and SEAO 10 mg/kg groups (F(5.48) = 23.13; *p* < 0.001) (Figure 12B).

## 3. Discussion

The supercritical extraction used in this study proved effective in obtaining an extract rich in spilanthol (25.91% of the total chromatographic area), which represents an advantage in both pharmacological standardization and process sustainability. Some studies in the literature have already mentioned that this technique is more efficient for obtaining *Acmella oleracea* flower extracts that are rich in spilanthol [16,19].

The extract obtained by supercritical CO_2_ extraction showed significant antiarrhythmic activity, with effects comparable to those of lidocaine, a well-established drug in the treatment of arrhythmias, especially ventricular arrhythmias.

The electrocardiograms of animals treated with SEAO at the three doses tested (10, 15, and 20 mg/kg) showed maintenance of sinus rhythm, confirming the hypothesis that the extract acts to modulate the heart’s electrical conduction. Tests conducted to evaluate the extract’s antiarrhythmic activity at the three SEAO doses showed a dose-dependent effect, most pronounced at the 20 mg/kg dose. This effect suggests an action similar to that of lidocaine, which blocks sodium channels in a dose-dependent manner [20]. Furthermore, the P-Q, QRS, and Q-T intervals were preserved or restored relative to the epinephrine group, indicating that SEAO acts protectively on the ventricular cardiac cycle and atrioventricular conduction.

The results of this study related to SEAO confirm previous studies indicating the extract’s cardioprotective action, attributed to its antioxidant capacity, anti-inflammatory action, and inhibition of the angiotensin-converting enzyme [21]. These effects are attributed to spilanthol, the extract’s main chemical marker and the bioactive compound responsible for most of the plant’s biological activities. The antiarrhythmic activity of SEAO obtained in this study represents an important advance in the search for new therapeutic products for the treatment of cardiovascular diseases, especially arrhythmias. However, more advanced studies are needed to clarify the molecular mechanisms involved, assess chronic toxicity, and ultimately evaluate the effects in human models.

Furthermore, the possibility of developing a new phytopharmaceutical from a plant native to the Amazon is of great relevance to the region’s bioeconomy and contributes to its social development.

## 4. Materials and Methods

### 4.1. Extraction and Characterization of the Extract

#### 4.1.1. Supercritical Extraction of *Acmella oleracea*

*Acmella oleracea* inflorescence extracts (SEAO) were obtained by supercritical extraction using CO_2_ as a solvent. The process was established based on the work developed by Silva et al. (2025) [22], which evaluated the selectivity of spilanthol in supercritical CO_2_ under different conditions. The process conditions of this work were 70 °C and 320 bar. These conditions provided the highest spilanthol content in the extract. The extractions were performed in a commercial Spe-ed™ SFE apparatus (Applied Separations, model 7071, Allentown, PA, USA), previously described by Batista et al. (2016) [23]. A mass of 10 g of raw material was used in a 100 mL extraction vessel. The static extraction time was 30 min, the dynamic extraction was 60 min, and the flow rate was 4.5 L/min. Extractions were performed in triplicate, and the results were expressed as % mass (db).

#### 4.1.2. Chromatographic Analysis of the Extract

The extract was subjected to transesterification and derivatization for subsequent quantification of the spilanthol content. Derivatization proceeded as follows: The solvent of the organic phase was evaporated, then 50 µL of N,O-bis(trimethylsilyl)trifluoroacetamide + 1% trimethylchlorosilane (BSTFA + 1% TMCS) was added. The solution was vortexed for 1 min and placed in a water bath with ultrasound-assisted stirring for 1 min at 30 °C. The BSTFA was evaporated, and 500 µL of the Hexane:CH_2_Cl_2_ (1:1) solution was added. Subsequently, the sample was transferred to a 2 mL glass vial with a cap and septum for subsequent GC-MS analysis.

Gas chromatography analysis was performed using a gas chromatograph (GC) (ThermoScientific Trace 1300, Waltham, MA, USA) coupled to a mass spectrometer (ThermoScientific MS-ISQ Single Quadrupole) with an AI 1310 autosampler, equipped with a ZB-5HT capillary column (30 m × 0.25 mm × 0.1 µm). Helium gas was used as a carrier at a flow rate of 1 mL/min. A 1.0 µL sample was injected in splitless mode. The injector operated at 220 °C and the oven temperature program started at 50 °C, rising to 200 °C (8 °C/min), holding for 1 min, increasing to 300 °C (15 °C/min), holding for 5 min, then rising again to 350 °C (15 °C/min) and holding for another 9 min. The MS-ISQ operated with an interface at 280 °C, an ionization source at 280 °C, mass range of 40–1000 Da. Electronic ionization was carried out at 70 eV. Substance identification was performed by comparing the mass spectrum with those of the commercial libraries NIST2011, WILEY2009 [24].

### 4.2. Biological Assays

#### 4.2.1. Animals

Twelve-week-old male Wistar rats (180–200 g) were kept in a controlled environment (25 ± 2 °C; 12 h light/dark cycle) with access to food and water *ad libitum*. A total of 90 animals were divided into two experiments: (I) evaluation of SEAO electrocardiographic activity (total of 54 animals); (II) evaluation of SEAO antiarrhythmic activity (total of 36 animals). All animals in all groups received 2.5 mg/kg i.p. of diazepam 10 min before all experiments. This work was approved by the Ethics Committee on Animal Experimentation of the Federal University of Pará (Brazil; license no. 001774) and followed the guidelines suggested by the Guide for the Care and Use of Laboratory Animals [4]. Every effort was made to reduce the number of animals and minimize animal suffering.

#### 4.2.2. Evaluation of the Electrocardiographic Activity of SEAO

To evaluate the electrocardiographic activity of SEAO, the animals were divided into six treatment groups with nine animals per group (n = 9). The negative control group (group 1) received 0.9% saline solution i.p. in an equivalent volume. The stimulant positive control group (group 2) received 0.25 mg/kg i.p. of epinephrine. The inhibitory positive control group (group 3) received 10 mg/kg i.p. of lidocaine. The groups treated with extracts (SEAO) received 10, 15, and 20 mg/kg i.p. in groups 4, 5, and 6, respectively. Electrocardiographic activity was measured as described by Muto et al. (2021) [25]. After 10 min of drug administration, electrocardiographic recordings were performed for 3 min for each animal, evaluating the following measurements: heart rate (bpm), amplitude (mV), R-R interval (ms), P-Q interval (ms), QRS duration (ms), Q-T interval (ms).

#### 4.2.3. Evaluation of the Antiarrhythmic Activity of SEAO

To evaluate the antiarrhythmic activity of SEAO, the 36 animals were divided into four groups (*n* = 9). Group 1 (positive control) received 10 mg/kg i.p. of lidocaine. Groups 2, 3, and 4 received 10, 15, and 20 mg/kg i.p. of SEAO, respectively. After 15 min of drug and extract administration, all animals received 0.25 mg/kg i.p. of epinephrine, and after 10 min, electrocardiographic recordings were measured for a period of 3 min, according to Muto et al. (2021) [25]. The following data were evaluated: heart rate (bpm), amplitude (mV), R-R interval (ms), P-Q interval (ms), QRS duration (ms), Q-T interval (ms).

#### 4.2.4. Analysis of Electrophysiological Data

A tool was built to analyze the obtained signals using Python version 2.7. The Numpy and Scipy libraries were used for mathematical processing applications, and the Matplolib library was used to generate the graphs. The graphical interface was developed using the PyQt4 library [26].

### 4.3. Statistical Analysis

Statistical analysis was performed using GraphPad Prism software (version 8.0). Parametric electrocardiographic (ECG) data were analyzed using ANOVA followed by Tukey’s test, and the results were expressed as the means ± standard deviation. Differences in the analyses were considered significant at *p*-values < 0.05.

## 5. Conclusions

The *Acmella oleracea* extract obtained by supercritical CO_2_ extraction showed a good yield of spilanthol for application in the arrhythmia model used in this study. SEAO exhibited dose-dependent antiarrhythmic action in the animal model tested, leading to the conclusion that SEAO has great potential for the development of a new phytopharmaceutical with antiarrhythmic action, especially in ventricular arrhythmias. Furthermore, we suggest that future studies should conduct pharmacokinetic, long-term, or chronic arrhythmia models to validate the efficacy of SEAO as an antiarrhythmic phytopharmaceutical.

## Figures and Tables

**Figure 1 plants-14-02848-f001:**
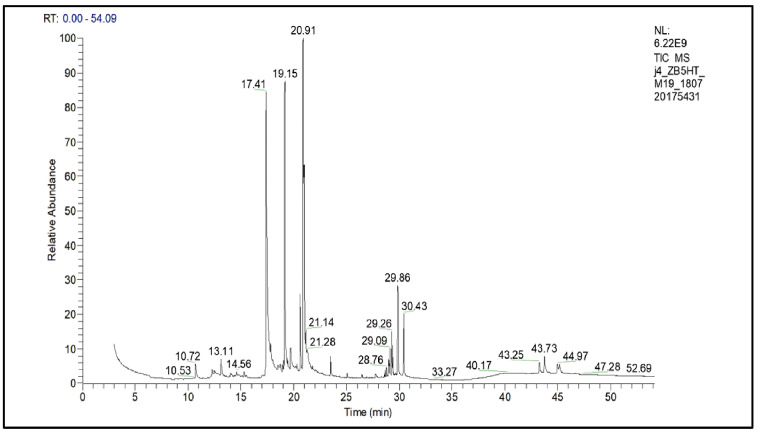
Chromatogram of the extract obtained by supercritical extraction.

**Figure 2 plants-14-02848-f002:**
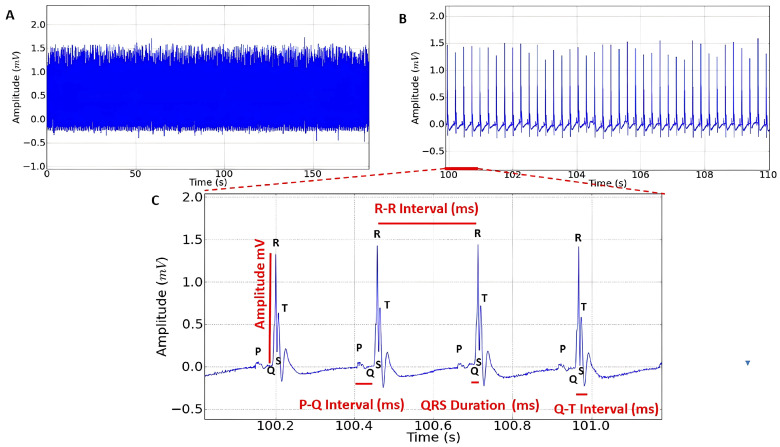
Control electrocardiogram in lead D-II in rats lasting 3 min. (**A**) Enlargement of the recording at 10 s, sinus rhythm observed; (**B**) enlargement of the recording at 1 s; (**C**) the red lines represent the intervals to be analyzed: amplitude (mV), R-R interval (ms), P-Q interval (ms), Q-T interval (ms), QRS complex duration (ms).

**Figure 3 plants-14-02848-f003:**
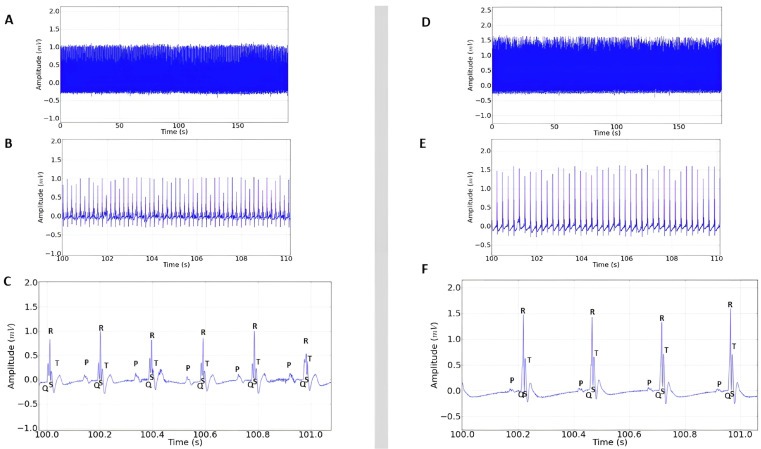
(**A**) Electrocardiogram in lead D-II in rats lasting 3 min after administration of 0.25 mg/kg i.p. of epinephrine; (**B**) 10 s magnification demonstrating sinus rhythm, extended recording period (100 to 110 s) after epinephrine administration; (**C**) 1 s magnification of the recording with the presence of P, QRS complex, and T triggers; (**D**) ECG recording represented in the tracing lasting 3 min after administration of 10 mg/kg i.p. of lidocaine; (**E**) 10 s magnification of the tracing (100 to 110 s); (**F**) 1 s amplitude demonstrating sinus rhythm after lidocaine administration.

**Figure 4 plants-14-02848-f004:**
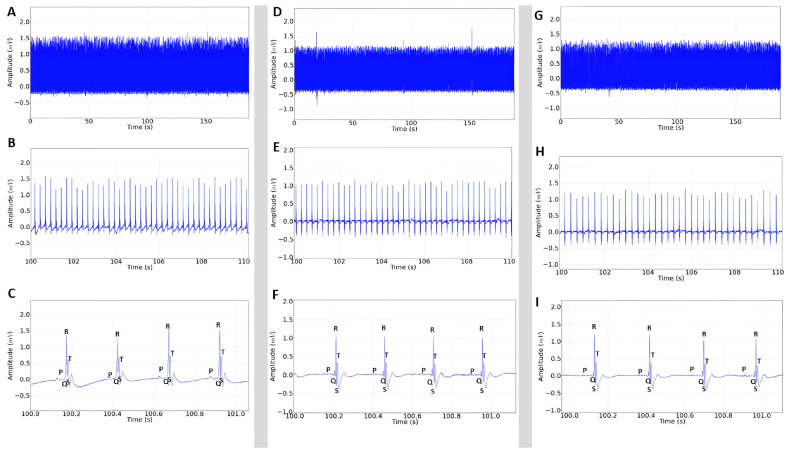
(**A**) Electrocardiogram in lead D-II in rats lasting 3 min after administration of 10 mg/kg i.p. of SEAO; (**B**) enlarged recording lasting 10 s; (**C**) enlarged recording in 1 s identifying the P, QRS complex, and T triggers; (**D**) recording represented in the 3 min tracing after administration of 15 mg/kg i.p. of SEAO; (**E**) enlargement of 10 s of the tracing; (**F**) enlargement of the tracing in 1 s, demonstrating cardiac triggers with sinus rhythm; (**G**) ECG tracing representing the use of SEAO at a dose of 20 mg/kg i.p. lasting 3 min; (**H**) enlarged recording in 10 s for rhythm assessment; (**I**) recording with 1 s enlargement demonstrating cardiac triggers.

**Figure 5 plants-14-02848-f005:**
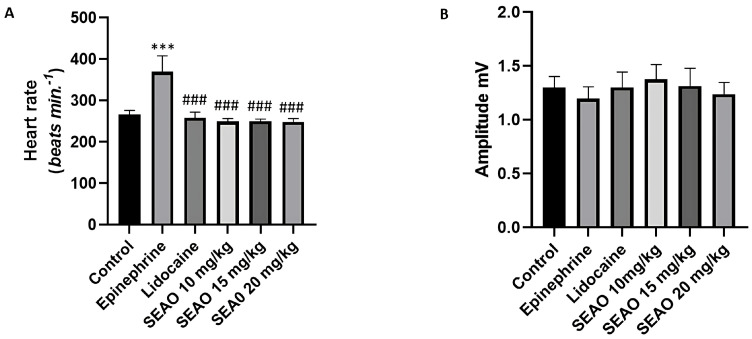
(**A**) Comparison of the mean heart rates (bpm) recorded in the groups to evaluate the electrophysiological changes after application of the tested drugs. (**B**) Evaluation of the mean amplitudes (mV) for animals treated with the drugs. (***) indicates a difference from the control group; (###) indicates a difference from the epinephrine group. [ANOVA and Tukey’s test (*p* < 0.05, *n* = 9)].

**Figure 6 plants-14-02848-f006:**
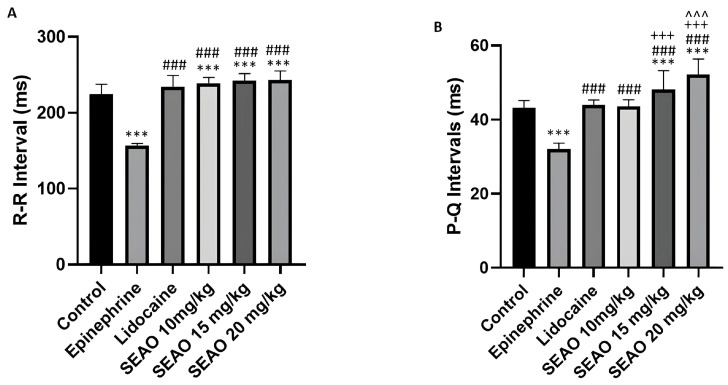
(**A**) Comparison of the mean R-R intervals (ms) recorded in the groups to evaluate the electrophysiological changes after application of the tested drugs. (**B**) Evaluation of the mean P-Q intervals (ms) for those treated with the drugs. (***) indicates difference from the control group; (###) indicates difference from the epinephrine group; (+++) indicates statistical difference from the lidocaine group, (^^^) indicates statistical difference from the SEAO 10 mg/kg group. [ANOVA and Tukey’s test (*p* < 0.05, *n* = 9)].

**Figure 7 plants-14-02848-f007:**
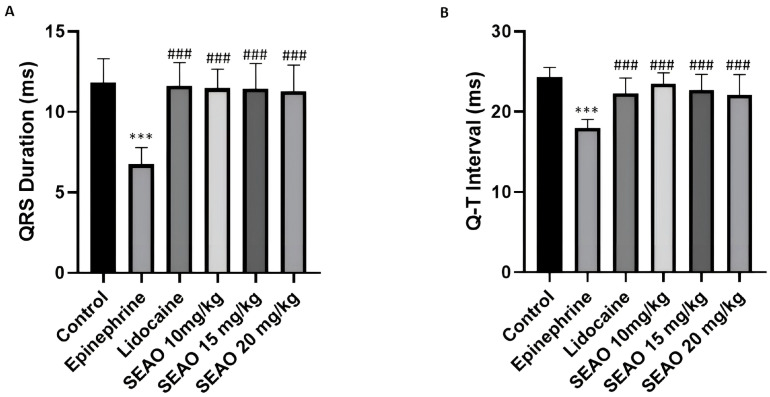
(**A**) Comparison of the mean QRS complex duration (ms) recorded in the groups after administration of the tested drugs. (**B**) Evaluation of the mean Q-T interval (ms) for the groups treated with the drugs. (***) indicates difference from the control group; (###) indicates difference from the epinephrine group. [ANOVA and Tukey’s test (*p* < 0.05, *n* = 9)].

**Figure 8 plants-14-02848-f008:**
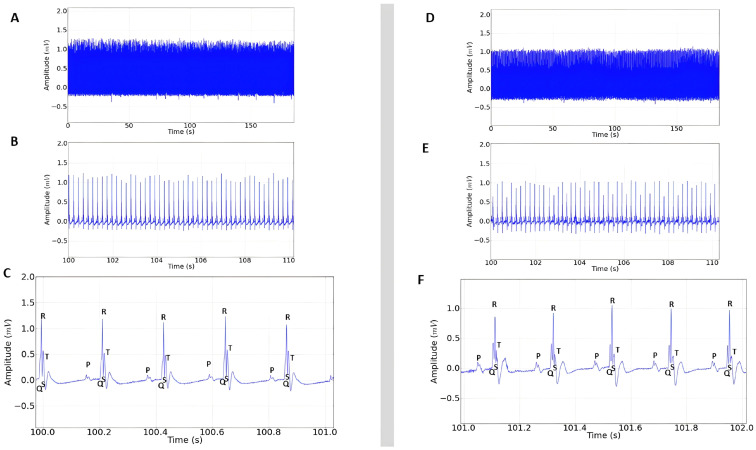
Electrocardiogram in lead D-II in rats lasting 3 min after administration of lidocaine 10 mg/kg i.p., followed by epinephrine 0.25 mg/kg i.p. (**A**); 10 s magnification demonstrating rhythm after epinephrine application (**B**); 1 s magnification of the recording with the presence of P, QRS complex, and T triggers (**C**); ECG recording represented in the tracing lasting 3 min after administration of 10 mg/kg i.p. of SEAO (**D**); 10 s magnification of the tracing (100 to 110 s) (**E**); 1 s amplitude demonstrating cardiac triggers on the ECG (**F**).

**Figure 9 plants-14-02848-f009:**
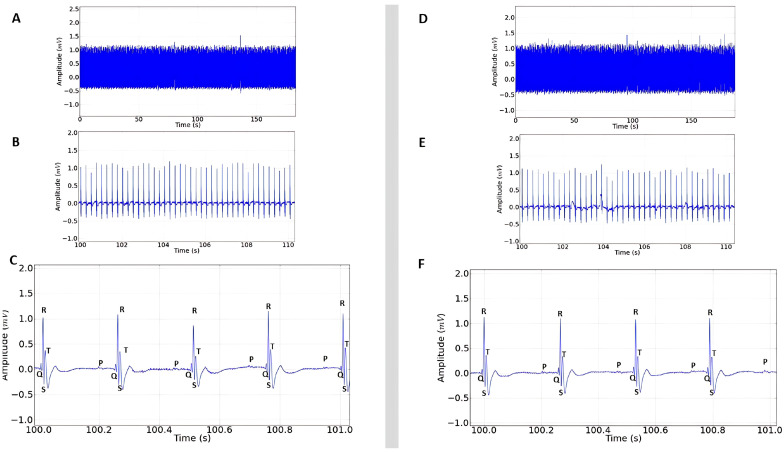
Electrocardiogram in lead D-II in rats lasting 3 min after administration of SEAO at a dose of 15 mg/kg i.p., followed by epinephrine 0.25 mg/kg i.p. (**A**); 10 s magnification (100–110 s) demonstrating cardiac rhythm after epinephrine application (**B**); 1 s magnification of the recording with a duration of 1 s indicating the P, QRS complex, and T triggers (**C**); ECG recording represented in the tracing lasting 3 min after administration of 20 mg/kg i.p. of SEAO (**D**); magnification of the recording with a duration of 10 s of the tracing (100 to 110 s) (**E**); amplitude of the recording with a duration of 1 s demonstrating the cardiac triggers of the ECG (**F**).

**Figure 10 plants-14-02848-f010:**
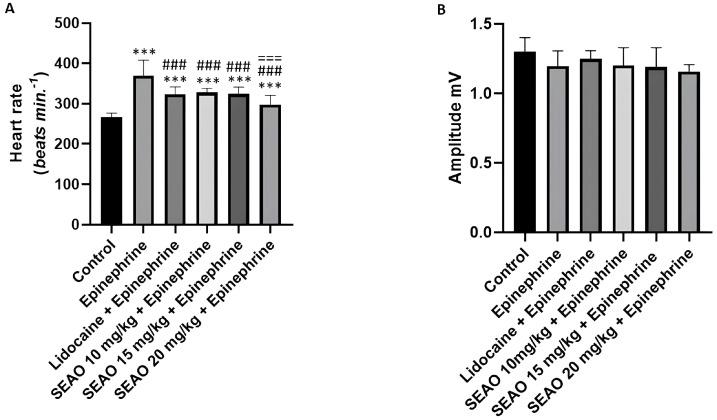
Comparison of the mean heart rate (bpm) recorded in the groups previously treated with lidocaine and SEAO to evaluate the electrophysiological changes after epinephrine application (**A**). Evaluation of the mean amplitude (mV) for animals previously treated with lidocaine and SEAO after epinephrine application (**B**). (***) indicates difference from the control group; (###) indicates difference from the epinephrine group; (===) indicates statistical difference from the SEAO 10 mg/kg + epinephrine group. [ANOVA and Tukey’s test (*p* < 0.05, *n* = 9)].

**Figure 11 plants-14-02848-f011:**
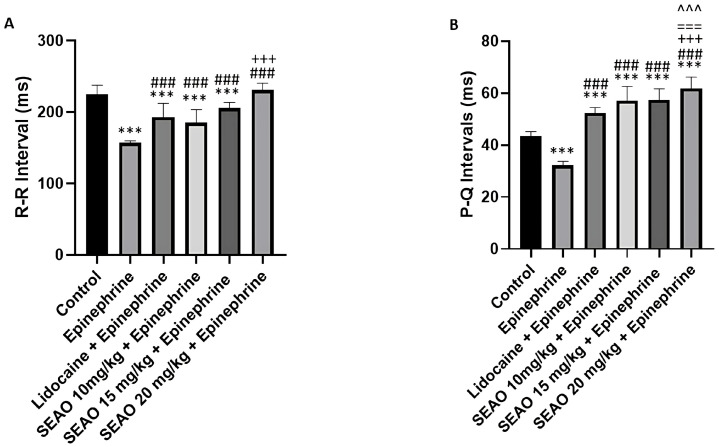
Comparison of the mean R-R interval (ms) recorded in the groups previously treated with lidocaine and SEAO to evaluate the electrophysiological changes after epinephrine application (**A**). Evaluation of the mean P-Q interval (ms) for animals previously treated with lidocaine and SEAO after epinephrine application (**B**). (***) indicates difference from the control group; (###) indicates difference from the epinephrine group; (+++) indicates statistical difference from the lidocaine + epinephrine group; (===) indicates statistical difference from the 10 mg/kg + epinephrine group; (^^^) indicates statistical difference from the SEAO 15 mg/kg + epinephrine group. [ANOVA and Tukey’s test (*p* < 0.05, n = 9)].

**Figure 12 plants-14-02848-f012:**
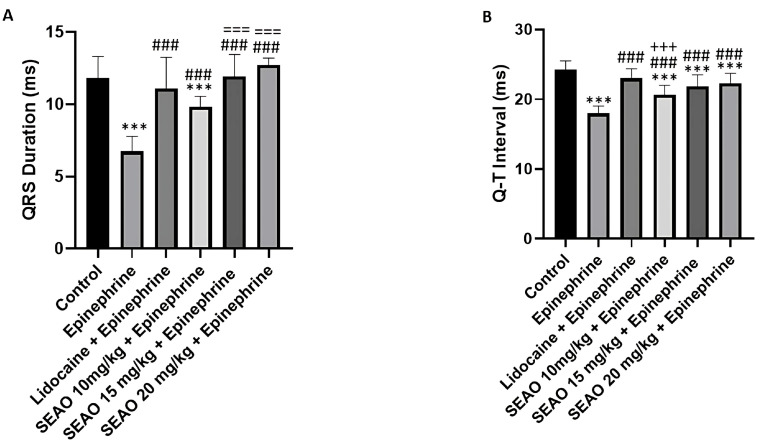
Comparison of the mean QRS complex duration (ms) recorded in the groups previously treated with lidocaine and SEAO to evaluate electrophysiological changes after epinephrine application (**A**). Evaluation of the mean Q-T interval (ms) for animals previously treated with lidocaine and SEAO after epinephrine application (**B**). (***) indicates difference from the control group; (###) indicates difference from the epinephrine group; (+++) indicates statistical difference from the lidocaine + epinephrine group; (===) indicates statistical difference from the SEAO 10 mg/kg + epinephrine group. [ANOVA and Tukey’s test (*p* < 0.05, *n* = 9)].

**Table 1 plants-14-02848-t001:** Electrocardiographic data of electrocardiographic activity.

Group	Frequency (bpm)	Amplitude (mV)	R-R (ms)	P-Q (ms)	QRS (ms)	Q-T (ms)
Negative Control	266.7	1.299	224.4	43.30	11.80	24.31
Epinephrine	369.3	1.196	156.6	32.14	6.744	17.99
Lidocaine	—	1.302	234.6	43.93	11.63	22.26
SEAO 10 mg/kg	249.3	1.376	239.1	43.53	—	23.48
SEAO 15 mg/kg	250.2	1.316	242.1	48.09	—	22.69
SEAO 20 mg/kg	248.9	1.238	243.2	52.24	—	22.11

**Table 2 plants-14-02848-t002:** Electrocardiographic data of antiarrhythmic activity.

Group	Frequency (bpm)	Amplitude (mV)	R-R (ms)	P-Q (ms)	QRS (ms)	Q-T (ms)
Control	266.7	1.299	224.4	43.30	11.80	24.31
Epinephrine	369.3	1.196	156.6	32.14	6.744	17.99
Lidocaine + Epinephrine	323.3	1.251	192.7	52.23	11.11	23.01
SEAO 10 mg/kg + Epinephrine	327.3	1.201	184.8	57.06	—	20.64
SEAO 15 mg/kg + Epinephrine	323.8	1.191	205.1	57.41	11.93	21.88
SEAO 20 mg/kg + Epinephrine	298.0	1.158	230.7	61.74	12.72	22.26

## Data Availability

The original contributions presented in this study are included in the article; further inquiries can be directed to the corresponding author.

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
