# Peer review of "Antiarrhythmic Effects of Supercritical Extract of Acmella oleracea in Rats: Electrophysiological Evidence and Cardioprotective Potential"

_plants, 2025, doi:10.3390/plants14182848_

Round 1
Reviewer 1 Report
Comments and Suggestions for Authors
The manuscript entitled ‘Antiarrhythmic Effects of Supercritical Extract of Acmella oleracea in Rats: Electrophysiological Evidence and Cardioprotective Potential' needs major revision.
THE MAIN NOTES:
1. The introduction to the article does not sufficiently explain why the authors chose to study the cardioprotective activity of the extract from the Acmella oleracea. There is no results were found by me in PubMed using the keywords 'Acmella AND Cardioprotection' or Acmella AND arrhythmia
2. The details of derivatization should be presented.
3. The dimension 'ml' should be written using the capital letter L everywhere - 'mL'. (see line 402).
4. The Latin name of the plant under study should be written everywhere using italics (see lines 77, 366, etc.)
5. The discussion section provides too few comparative data with literature sources (only 4). This should be corrected by comparing data on other plant species with cardioprotective activity or at least on other species of the genus Acmella - https://pubmed.ncbi.nlm.nih.gov/?term=Acmella&size=50
6. The Conclusions section should be expanded to reflect the main achievements of this work.
7. And finally, analyzing the iThenticate report, it could be concluded that the authors should increase the percentage of originality of the text, as it has a Percent match of 31%
Reviewer 2 Report
Comments and Suggestions for Authors
This manuscript presents a robust preclinical investigation into the antiarrhythmic effects of a supercritical COâ‚‚ extract of Acmella oleracea (SEAO) in Wistar rats. The study is timely and relevant, addressing the urgent need for safer and natural antiarrhythmic agents. The work demonstrates scientific merit, methodological rigor, and originality. The supercritical extraction process and quantification of spilanthol (25.91%) are particularly well-executed. The findings show that SEAO has dose-dependent antiarrhythmic effects comparable to lidocaine, with preserved ECG parameters across key intervals (R-R, P-Q, QRS, QT).
Major Comments:
- Translational Considerations:
- The discussion would benefit from further elaboration on the clinical translatability of SEAO, including bioavailability, metabolism, and safety in humans.
- Consider acknowledging the need for pharmacokinetics, long-term toxicity, or chronic arrhythmia models in future studies. - Limitations:
- The study lacks explicit discussion on potential limitations (e.g., sample size, single species, acute vs. chronic treatment).
- Include a brief paragraph reflecting on these limitations and how they can be addressed in follow-up studies. - Graphical Summary/Table:
- The inclusion of a table summarizing the ECG parameter differences across treatment groups would enhance readability and serve as a quick-reference tool for readers.
Minor Comments:
- Formatting:
- The manuscript includes inline line numbers (e.g., “53”, “60”), which should be removed prior to final submission.
- Figure labels (e.g., "Figure 5 – (A)...") should be made consistent in font and formatting. - Typographical and Grammatical Issues:
- Check for minor typos throughout (e.g., “cardiac triggerings” → “cardiac signals”).
- Standardize capitalization (e.g., “Electrocardiographic” vs “electrocardiographic”). - Chemical Data:
- Clarify if 25.91% spilanthol refers to weight %, chromatographic area %, or another metric. Precision in chemical quantification enhances reproducibility. - Citation Formatting:
- Ensure all in-text citations match journal formatting requirements (e.g., [1], [2]) and are linked properly to the reference list.
Conclusion:
This is a high-quality manuscript that advances the pharmacological understanding of Acmella oleracea and supports its potential as a natural antiarrhythmic agent. The experimental design is solid, the results are convincing, and the writing is generally clear. With minor revisions to formatting and discussion depth, the manuscript is suitable for publication.
Round 2
Reviewer 1 Report
Comments and Suggestions for Authors
This manuscript has undergone significant revisions by the authors and can now be accepted for publication.